# Integrated Approach to Chronic Pain—The Role of Psychosocial Factors and Multidisciplinary Treatment: A Narrative Review

**DOI:** 10.3390/ijerph21091135

**Published:** 2024-08-28

**Authors:** Irena Kovačević, Jadranka Pavić, Biljana Filipović, Štefanija Ozimec Vulinec, Boris Ilić, Davorina Petek

**Affiliations:** 1Department of Nursing, University of Applied Health Sciences, 10000 Zagreb, Croatia; jadranka.pavic@zvu.hr (J.P.); biljana.filipovic@zvu.hr (B.F.); stefanija.ozimec-vulinec@zvu.hr (Š.O.V.); boris.ilic@zvu.hr (B.I.); 2Department of Nursing, Faculty of Health Studies, University of Rijeka, 51000 Rijeka, Croatia; 3Department of Family Medicine, Faculty of Medicine, University of Ljubljana, Poljanski Nasip 58, 1000 Ljubljana, Slovenia; davorina.petek@gmail.com

**Keywords:** chronic non-malignant pain, narrative review, biopsychosocial model, multidisciplinary approach, psychological factors, social factors

## Abstract

Background: Chronic non-malignant pain represents a growing global public health priority. Chronic pain is multifactorial, with numerous biological, psychological, and social factors contributing to this pain syndrome. It affects not only the patients, impairing their quality of life, but also their family and social environment. Chronic pain is a diagnosis and requires effective and sustainable treatment strategies. Objective: Our aim was to critically review the available evidence on the importance of different approaches in treating patients with chronic non-malignant pain, emphasizing the effectiveness of integrating psychological and social factors within a multidisciplinary framework. Methods: This was a non-systematic narrative review of the basic and recent literature analyzing approaches to the treatment of chronic non-malignant pain. The inclusion criteria for the papers were chronic non-malignant pain, treatment approach, review, and original research papers published in English in the last five years (PubMed search), and the basic literature was selected from the references of new papers according to the knowledge and experience of the authors. Results: This literature review included 120 papers, of which 83 were basic, and 37 were new, published in the last 5 years (2018–2023). The results show that both the basic and newly published literature advocate for a biopsychosocial approach to treating chronic pain. Conclusions: New findings, compared to the earlier literature, indicate a new classification of chronic pain into primary and secondary. Chronic pain should be approached with a biopsychosocial model within a multidisciplinary treatment framework. This model addresses the complex interplay of biological, psychological, and social factors, offering a holistic strategy for effective pain management.

## 1. Introduction

Chronic pain is a multifactorial phenomenon, encompassing numerous biological, psychological, and social factors that contribute to this pain syndrome [1]. Patients suffering from chronic pain often face misunderstanding, rejection, and stigmatization, which profoundly shape their path through suffering [2]. The identification of the psychological and behavioral health factors associated with chronic pain, as well as the challenges and opportunities for integrating multidisciplinary care into chronic pain management, is a crucial element in optimizing clinical outcomes [3], as Chandler et al. emphasized [4]. Therefore, chronic pain should be given more attention as a global health priority since adequate pain management is a human right, and every healthcare system has to provide it [5,6]. In the professional literature, we can find several definitions, but, generally, chronic pain is defined in terms of its duration and recurrence, being pain with a duration that exceeds the expected recovery period after an injury or surgical procedure and lasts more than three months [7]. According to Vellucci, chronic pain is pain that lasts at least three months and is characterized by recurring and/or continuous episodes of pain [8]. In its guidelines for pain assessment and treatment, the Wisconsin Pain Treatment Task Force states that chronic pain persists despite the resolution of its cause, thus being independent of an organic precipitating factor [9]. According to Wall and Melzac, chronic non-malignant pain is generally considered to last longer than 6 months, is caused by non-life-threatening conditions, has not responded to currently available treatment methods, and can persist for a lifetime [10].

In its guidelines for managing chronic pain issued in 2009, the Institute for Clinical Systems Improvement distinguishes four categories of chronic pain [11]:Neuropathic pain [either peripheral, including postherpetic neuralgia, diabetic neuropathy; or central, including pain after stroke or multiple sclerosis];Musculoskeletal pain [e.g., back pain, myofascial pain syndrome, ankle pain];Inflammatory pain [e.g., inflammatory arthropathies, infection];Mechanical/compression pain [e.g., kidney stones, visceral pain due to the expansion of tumor masses].

According to the new International Classification of Diseases (ICD-11), there are seven main categories of chronic pain:5.Chronic primary pain: Chronic primary pain is chronic pain in one or more anatomical regions characterized by significant emotional distress (anxiety, anger/frustration, or depressed mood) or functional disability (impairment of activities of daily living and reduced participation in social roles). Chronic primary pain is considered a health problem, distinct from the six categories of chronic secondary pain. In chronic secondary pain, the pain is considered a symptom of the underlying disease. The categories for chronic secondary pain are as follows [12]:6.Chronic cancer-related pain;7.Chronic post-surgical or post-traumatic pain;8.Chronic secondary musculoskeletal pain;9.Chronic secondary visceral pain;10.Chronic neuropathic pain;11.Chronic secondary headache or orofacial pain.

The ICD-11 classification is suitable for coding both chronic primary and chronic secondary pain, providing a more precise and less ambiguous presentation of chronic pain conditions in health statistics [13].

In 2020, the International Association for the Study of Pain (IASP) adopted a revised definition of pain: “an unpleasant sensory and emotional experience associated with, or resembling that associated with actual or potential tissue damage” [14]. The revised definition, highlighting pain as a personal experience influenced by biological, psychological, and social factors, emphasizes the subjective nature of pain, which does not always have to be directly related to physical damage but can be influenced by various biological, psychological, and social factors [14].

The definition of pain has profound implications for clinical practice, encouraging a holistic, individualized, and interdisciplinary approach to treatment. Understanding and applying this definition can lead to improvements in diagnosis, treatment, and ultimately, outcomes for patients experiencing pain.

Various models for addressing chronic pain have been developed, each offering unique approaches to treatment. The biomedical model focuses on the physical and biological aspects of pain, often emphasizing pharmacological interventions and surgical treatments. In contrast, the biopsychosocial model considers the complex interaction of biological, psychological, and social factors, advocating for a more holistic approach to pain management. This model supports the use of multidisciplinary treatment programs that integrate medical, psychological, and social support [1,2].

The treatment options for chronic pain are diverse and may include pharmacotherapy, physical therapy, psychological interventions, and complementary therapies such as acupuncture and massage [3,4]. Recent advances also highlight the importance of cognitive–behavioral therapy (CBT) and other psychosocial interventions in the effective management of chronic pain [1,4].

The main objective of this study was to critically review the available evidence on the importance of different approaches in treating patients with chronic non-malignant pain, emphasizing the effectiveness of the biopsychosocial model and multidisciplinary treatment programs.

## 2. Materials and Methods

A non-systematic review of the foundational and new literature on the approach to the treatment of chronic non-malignant pain was conducted. The inclusion criteria for the recently published papers were chronic non-malignant pain, treatment approach, original research and review papers, published in English, in the last five years (PubMed database search). The foundational papers were selected from the references of new papers according to the authors’ knowledge and experience. This literature review included 120 papers, of which 83 are foundational, and 37 are new papers published in the last 5 years [2018–2023]. Studies were excluded if they did not focus on chronic non-malignant pain; were opinion pieces, editorials, or letters to the editor without original research data; did not include a multidisciplinary treatment approach.

We used MeSH terms “Chronic Pain”, “Pain Management”, and “Models, Biopsychosocial”, and combined them with free text terms like “non-malignant pain” and “treatment approaches”. Boolean operators were used to refine the search. For instance, searches included combinations like “Chronic Pain AND Non-malignant pain Management AND Models, Biopsychosocial”. This search strategy allowed us to capture a broad range of relevant articles.

## 3. Results and Discussion

This section presents our findings, organized into several key categories identified in the literature. First, the epidemiology of chronic pain is presented, including its prevalence and impact on quality of life. Next, factors associated with the onset of chronic pain are discussed, including demographic and social characteristics. Following this, patients’ attitudes, beliefs, and expectations regarding chronic pain and its treatment are analyzed. Lastly, a multidisciplinary approach to chronic pain treatment is described, along with the role of the biopsychosocial model and social support in managing chronic pain. Each of these categories provides insights into different aspects of chronic pain, offering a comprehensive understanding of this complex issue.

### 3.1. Epidemiology of Chronic Pain

#### 3.1.1. Prevalence of Chronic Pain

European data indicate that moderate- and high-intensity chronic pain with a serious impact on daily life, social status, and working life occurs in 19% of the adult European population [15]. Patients suffering from chronic pain consume almost twice as many healthcare resources as the general population [16]. In the United States, 20.4% of adults suffer from chronic pain and 7.4% from severe chronic pain, with women, non-Hispanic whites, and people over the age of 65 being the most affected [17].

#### 3.1.2. Chronic Lower Back Pain

Chronic lower back pain (LBP) exemplifies the significant health and social impact of chronic pain. It is one of the most prevalent global health issues and the most common musculoskeletal problem worldwide [18,19,20,21]. As of 2020, it was estimated that 619 million people globally suffered from LBP, with projections suggesting that this number will rise to 843 million by 2050. Non-specific LBP, which accounts for approximately 90% of cases, imposes a significant burden on quality of life and productivity [22,23]. In 2017, a global study found that the prevalence of LBP was 7.50%, and, in 13 out of 21 world regions, it was the most common cause of years lived with disability (YLD). Western Europe, in particular, reported the highest YLDs due to LBP, underscoring the condition’s severe impact on work capacity and overall well-being [24]. Significantly, 38.8% of YLDs were linked to modifiable risk factors, including occupational hazards, smoking, and high BMI, highlighting the potential for prevention and the need for enhanced management strategies [22]. Given its widespread impact, the effective management and prevention of LBP are critical for reducing the global burden of chronic pain.

#### 3.1.3. Economic Costs of Chronic Pain

The extent of chronic pain management costs is illustrated by the fact that 12% of all prescription drugs are associated with chronic pain management, with more than USD 100 billion in direct and indirect costs [25]. According to a study by Pico and Clark, pain-related costs (direct costs and loss of earnings) in the US exceed the total cost of treating cancer, heart disease, and diabetes [26]. One-fifth of respondents believed that their family doctor did not consider their pain a problem. As many as 40% of the respondents reported that doctors preferred to treat their disease, i.e., their diagnosis, rather than their pain [15]. It can be said that chronic pain is becoming a global public health problem with an increasing prevalence despite numerous studies, an increasing number of pharmacological and non-pharmacological procedures, and the establishment of pain management clinics [27].

### 3.2. Factors Associated with the Onset of Chronic Pain

Numerous studies indicate a relationship between chronic pain and various demographic, socioeconomic, and lifestyle factors. These factors can significantly influence both the onset and severity of chronic pain. The key risk factors associated with chronic pain are clearly outlined in Table 1, providing a structured overview of these factors.

#### 3.2.1. Detailed Analysis of Specific Risk Factors

##### Sex

Research shows that women are at an increased risk of chronic pain compared to men. Women tend to report their pain more frequently, have a higher sensitivity to pain stimuli, and often experience pain conditions such as fibromyalgia more acutely. This difference is partly due to hormonal, genetic, and psychosocial factors. For example, women are more likely to tolerate pain when they focus on it and reinterpret the sensation, whereas men might respond better to distraction techniques [28,29].

##### Age

Older age is strongly associated with the development of chronic pain. As individuals age, they are more likely to experience degenerative conditions such as osteoarthritis and spinal stenosis, which contribute to chronic pain. This increased risk is due to the natural wear and tear of the musculoskeletal system over time [29].

##### Socioeconomic Status

Socioeconomic status (SES) is a significant determinant of chronic pain. Individuals with lower SES are more likely to experience chronic pain due to factors such as limited access to healthcare, lower levels of education, and increased exposure to physically demanding or hazardous work environments. These factors not only increase the likelihood of pain but also reduce the individual’s ability to effectively manage it, leading to serious physical and psychosocial disabilities [30,31,32].

##### Obesity

Obesity is another major risk factor for chronic pain, particularly in older adults. Studies have shown that individuals with moderate obesity are twice as likely to develop chronic pain compared to those with normal weight, and those with severe obesity are more than four times as likely to suffer from chronic pain. This association is particularly evident in conditions such as osteoarthritis, where excess weight puts additional strain on the joints [30].

##### Additional Risk Factors

Geographic and cultural background: Cultural perceptions and geographic factors can significantly influence how pain is reported and managed. Cultural beliefs about pain, healthcare access, and regional healthcare practices all play a role in chronic pain outcomes [31].

Employment status: Job instability or unemployment is correlated with higher levels of chronic pain, likely due to the stress and financial insecurity that accompany these conditions. Chronic pain can also lead to absenteeism and the need for social welfare support, further exacerbating the individual’s financial and social situation [32,34].

History of abuse: A history of physical or emotional abuse is linked to an increased likelihood of developing chronic pain, as trauma can have long-lasting effects on the nervous system and pain perception [33].

Negative interpersonal relationships: Poor social support and strained interpersonal relationships can contribute to the persistence and exacerbation of chronic pain, highlighting the importance of addressing these factors in pain management strategies [34,35,36].

### 3.3. Attitudes, Beliefs, and Expectations of Patients

Culture [37,38], attitudes [39,40], beliefs, and religion [41,42] also play an important role in chronic pain. In general, pain-related attitudes and beliefs are important predictors in identifying those likely to develop long-term pain [40,43]. The results of the 2021 Najem study suggest the possibility that spirituality, hope/optimism, as well as spiritual/religious beliefs may be important sources of meaning, giving individuals a sense of purpose, and that spirituality may be a useful source of psychological adjustment and promote the use of adaptive coping strategies [44]. Patient expectations can also be significantly affected by psychological distress, including negative mood, lack of positive attitude, and prolongation of disability. Therefore, the psychological impact on pain should be carefully considered, as it can significantly influence pain intensity, patient expectations, and physical function [45]. Research shows that psychosocial interventions could contribute to improved psychological well-being, also resulting in an improvement in the patient’s self-efficacy against pain [46].

Since the mid-1980s, numerous researchers have advocated for the routine assessment of patients’ attitudes, beliefs, and expectations regarding pain and its treatment. Strong et al. and Slater et al. hypothesized that an assessment of attitudes should be included before starting treatment as a part of a multidimensional assessment [47,48]. The above information, including the ability to cope with pain, is valuable for treatment planning and as an indicator of expected treatment outcomes [49,50,51,52,53]. Attitudes towards pain, as defined by Fishbein et al., encompass both cognitive and affective components, representing the degree of emotion or affect towards a certain object, as well as the patient’s understanding of pain and what it represents for them [52,54].

According to Spinhoven et al., there are two types of beliefs. One of them is attributions, and the other is expectations. Attributions refer to interpretation in terms of importance and potential danger, while expectations refer to thoughts related to expected consequences and include thinking about a person’s ability to control pain and the effectiveness of those efforts [55]. The patient’s belief about the causes of pain and the expected effect of the treatment can also influence the decision to accept the treatment and its likely outcome [56,57]. Beliefs differ from attitudes to the extent that beliefs include information about an object [54].

Foster et al. conducted a study on the relationship between the perception of pain experience and clinical outcomes. The results of the study revealed that subjects with favorable clinical outcomes perceived less serious consequences, reported fewer emotional reactions such as fear and anger, and had fewer symptoms and a stronger perception of control over their problem [58]. Patient’s coping strategies, from passive to active, resulted in reduced pain perception and increased satisfaction with treatment results [59]. Pain-related attitudes and beliefs play a significant role in how individuals perceive and manage chronic pain. Understanding these cognitive aspects is crucial for effective pain management. Several validated tools are available for assessing these attitudes and beliefs. The first tool developed was the Pain Information and Belief Questionnaire (PIBQ), designed to gather information about patients’ beliefs and understanding of their pain [49,53]. Following this, the Pain Impairment Relationship Scale (PAIRS) was introduced to measure the degree to which patients with chronic pain believe that pain impacts their performance [51]. These tools help clinicians identify maladaptive beliefs and provide a basis for cognitive–behavioral interventions aimed at improving pain management outcomes.

According to research by Symonds et al., the negative attitudes and beliefs of people with chronic pain may also affect the achievement of the desired treatment outcomes [60]. The perception of pain and the fear of the duration of the disease can be predictors for anticipating the patient’s recovery and return to work [61,62,63]. May concluded that changing the patient’s pain-related attitudes and beliefs could speed up recovery and return to daily activities [64]. According to Darlow et al., the most prevalent fear among patients is that chronic pain will affect their work performance and that physical activity will generate more negative consequences related to pain [65]. Hanney et al. and Linton et al. hypothesized that negative beliefs among patients with chronic pain may even worsen pain, which in turn leads to functional limitations and chronic pain patterns [66,67]. The training of healthcare professionals should enable them to better understand and accept the patient’s knowledge, certainly including their health-related beliefs and expectations [68].

### 3.4. Biopsychosocial Model of Treatment and Social Support

The biopsychosocial model, proposed by Engel as an alternative to the biomedical approach, offers a comprehensive framework for understanding illness and health by incorporating biological, psychological, and social factors [69,70]. This model is particularly relevant in the context of chronic pain, as it accounts for the complexity of the individual pain experience, which is influenced by sensory, cognitive–affective, and interpersonal factors [71,72]. Pain and nociception are distinct phenomena; pain cannot be fully understood by sensory neuron activity alone [14,73].

#### 3.4.1. Application of the Biopsychosocial Model in Chronic Pain Management

The biopsychosocial model is considered essential for effective pain treatment. It provides a framework for multidisciplinary biopsychosocial rehabilitation, which integrates physical training and patient education (biological component), cognitive–behavioral treatment (psychological component), and discussions about issues in the work and social environments (social component) [74,75,76,77,78]. This multidisciplinary strategy, which includes non-pharmacological approaches, is designed to empower patients with chronic pain to self-manage their symptoms [79,80].

#### 3.4.2. Challenges and Benefits of the Biopsychosocial Approach

The biopsychosocial model encourages a holistic view of pain, considering all aspects of the patient’s nature, from physiology to social relationships and mental awareness [81,82]. Traditional biomedical treatment methods often fail to adequately manage chronic pain and may contribute to further disability [83,84,85]. In contrast, the biopsychosocial approach emphasizes a partnership between the patient and healthcare providers, fostering patient autonomy and reducing the impact of chronic pain on daily life [86,87,88]. The biopsychosocial model has been proven to be a valuable framework for chronic pain management, especially in improving patient outcomes through a comprehensive and integrated approach [89,90]. Despite the advancements in this field, there remains a need for greater emphasis on psychological and social factors in treatment plans [91,92]. The COVID-19 pandemic has highlighted the importance of biopsychosocial interventions, including telerehabilitation, in maintaining patient care under challenging circumstances [93,94,95]. As research continues to evolve, the integration of traditional and innovative approaches will be crucial in addressing the multifaceted nature of chronic pain [96,97].

#### 3.4.3. Innovative Approaches and Future Directions

In addition to traditional treatments such as pharmacotherapy, psychotherapy, and physical therapy, innovative approaches like virtual reality (VR) are emerging as promising tools in chronic pain management. VR allows patients to engage in immersive environments that can help reduce pain perception and improve associated conditions such as anxiety and depression [98]. Although preliminary findings are promising, further research is needed to validate the effectiveness of VR in clinical settings.

### 3.5. A Multidisciplinary Approach to the Treatment of Chronic Pain

#### 3.5.1. Introduction to Multidisciplinary Treatment

The effective treatment of chronic pain requires a multidisciplinary and cognitive-behavioral approach [99]. This approach integrates various treatment modalities, ensuring that patients receive comprehensive care tailored to their specific needs. Successful outcomes involve a personalized, step-by-step approach using pharmacotherapy, psychotherapy, integrative treatments, and interventional procedures [100].

#### 3.5.2. Benefits of Multidisciplinary Care

Patients receiving multidisciplinary care benefit from early diagnosis and treatment, which is crucial to managing the underlying conditions associated with chronic pain. The involvement of health professionals from various specialties allows patients to choose between pharmacological and non-pharmacological treatments, resulting in more effective and individualized care [101]. Research indicates that simultaneous and early intervention addressing physiological, psychological, and social aspects can reduce pain perception, increase psychosocial well-being, and save costs for society [102]. The effects of a multidisciplinary approach extend beyond pain relief to include increased muscle activity and strength, suppression of pain-related behaviors, reduced reliance on certain drugs, decreased depression and social isolation, and return to work [101,103]. However, treatments focusing mainly on pharmacotherapy and interventional procedures result in the increased use and abuse of opioids [104]. Ignoring the role of behavioral and psychological factors in patient care is a disservice to patients and a missed opportunity for better fiscal management of resources [105].

#### 3.5.3. Implementation of Multidisciplinary Teams

To simplify access to multidisciplinary pain treatment, it is essential to include psychosocial interventions alongside restorative, behavioral, complementary, and integrative therapies [106]. Multidisciplinary pain centers should be equipped to treat any type of pain, with healthcare professionals possessing adequate knowledge of clinical practices relevant to chronic pain treatment and being informed of all relevant guidelines. This ensures up-to-date, evidence-based, and safe treatment, with a coordinator supervising medical services to maintain high standards [107,108]. Regular communication among multidisciplinary team members about specific patients and overall progress is crucial to ensuring continuity of care, avoiding duplicate medical testing, and identifying treatment failures early [107,109].

#### 3.5.4. Impact on Patient Outcomes

Experts agree on the importance of a multifaceted approach to chronic pain management that includes interdisciplinary communication and coordination. Studies show that patients with complete documentation of their problems, including chronic pain, receive better follow-up care [110]. Falkham et al. highlighted the positive effects of interdisciplinary multimodal programs in primary healthcare on reducing pain intensity; improving physical and emotional function, physical activity, and health-related quality of life; as well as reducing sick leave, both short-term and long-term [111]. Similarly, Connell suggested that interdisciplinary interventions involving teamwork improve pain treatment outcomes compared to usual care [112].

#### 3.5.5. Structure of Multidisciplinary Teams

The structure of multidisciplinary teams varies but usually includes three doctors (e.g., primary care physician, anesthesiologist, and psychiatrist) and other health professionals (e.g., psychologist, physiotherapist, and nurse) [113]. Other team members may include neurologists, orthopedists, neurosurgeons, and rehabilitation therapists. Multidisciplinary teams can also provide consultative services, making recommendations through case reviews and discussions with primary care providers for psychosocially complex patients suffering from pain and addiction [114].

#### 3.5.6. Early Intervention and Primary Healthcare

Early treatment of low back pain symptoms at the primary healthcare level, rather than in hospitals, can reduce disability and pain intensity and speed up return to work [115]. The comprehensive multidisciplinary treatment of chronic non-malignant pain, emphasizing various strategies and specialist treatments performed by a multidisciplinary team, is clinically and economically effective compared to non-multidisciplinary treatment or usual healthcare [116,117,118]. Early access to pain management, minimizing late diagnosis, achieving effective treatment, and avoiding costly complications, is the key advantage of this approach. This can significantly improve patients’ quality of life and the sustainability of healthcare systems [119]. Excluding patients with complex chronic pain from primary healthcare, as suggested by Dassieu et al., would improve the availability of multidisciplinary treatment for these patients [120].

The steps and decision points in the process of treating chronic non-malignant pain are outlined in Figure 1. Different approaches to chronic non-malignant pain management are shown in Figure 2.

## 4. Summary and Clinical Implications

Treatment of chronic pain requires multidisciplinary cooperation and the continuous monitoring of therapy effectiveness, with the more active involvement of primary health care as the first level at which patients seek help. Considering biological, psychological, and sociological factors, the biopsychosocial model of treatment becomes necessary in the treatment of chronic pain. Social support stands out as a powerful factor in better health because it reduces the perception of stressful and harmful events. Support from family, friends, and even doctors contributes to the patient’s activation, better pain tolerance, and reduced pain intensity. Since the outcome of the treatment of chronic non-malignant pain is influenced by many biopsychosocial factors other than medical ones, the treatment approach should be adapted to these differences. For a successful outcome in the treatment of patients with chronic non-malignant pain, healthcare professionals should consider a broader range of factors based on the biopsychosocial model.

It can be concluded that the treatment of chronic pain requires an integrated approach that includes the biological, psychological, and social aspects of the disease. This paper highlights the multifactorial nature of pain and the advantages of the biopsychosocial model, which, together with a multidisciplinary approach, provides a comprehensive framework for understanding and treating chronic pain. Early recognition and intervention, along with patient education and integration of psychosocial interventions, are critical in successful treatment.

## Figures and Tables

**Figure 1 ijerph-21-01135-f001:**
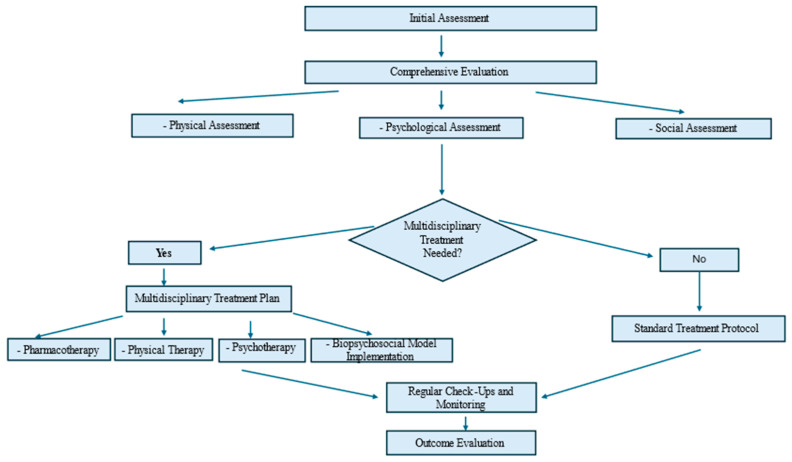
Flow chart—steps and decision points in the process of treating chronic non-malignant pain.

**Figure 2 ijerph-21-01135-f002:**
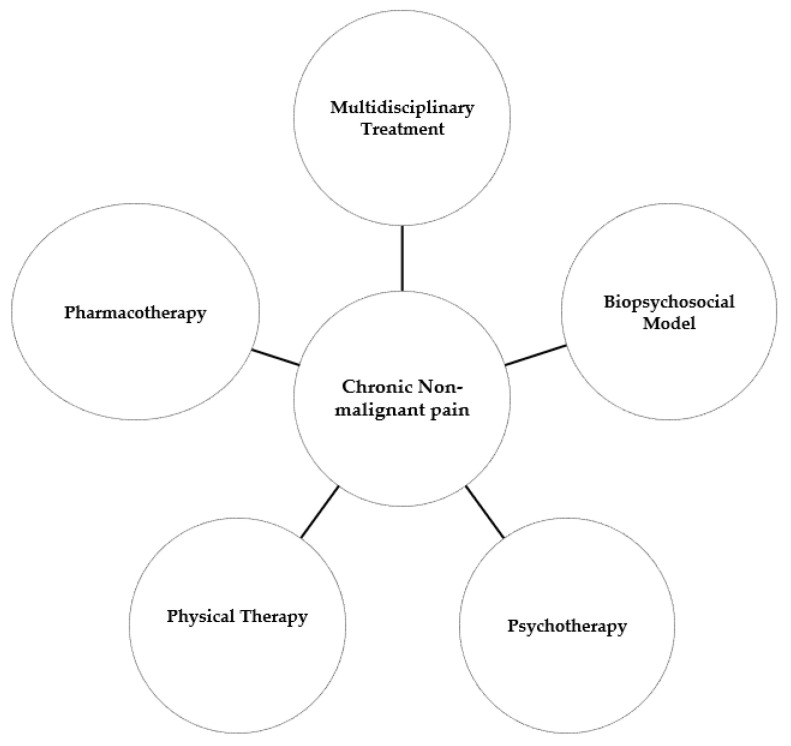
Chronic non-malignant pain treatment approaches.

**Table 1 ijerph-21-01135-t001:** Key risk factors for chronic pain and associated studies.

Risk factor	Study	Findings
Gender (female)	[28]	Women are more likely to report chronic pain and have higher pain sensitivity.
Older age	[29]	Older individuals are at a higher risk of developing chronic pain.
Socioeconomic status	[30]	Lower socioeconomic status is associated with a higher prevalence of chronic pain.
Geographic and cultural background	[31]	Cultural factors influence the perception and reporting of pain.
Employment status	[32]	Unemployment or unstable employment is associated with higher pain prevalence.
History of Abuse	[33]	A history of physical or emotional abuse increases the likelihood of chronic pain.
Negative Interpersonal Relationships	[34]	Poor social support and relationships contribute to the persistence of pain.
Obesity	[30]	Obesity significantly increases the likelihood of developing chronic pain.

## Data Availability

Not applicable.

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
