# Peer review of "Integrated Approach to Chronic Pain—The Role of Psychosocial Factors and Multidisciplinary Treatment: A Narrative Review"

_ijerph, 2024, doi:10.3390/ijerph21091135_

Round 1

Reviewer 1 Report

Comments and Suggestions for Authors

great revisions; thank you. 

I noticed this sentence at the end of the materials and methods section, which seems to be unnecessary: "We have clarified this methodology in the revised Methods section of the manuscript"

Otherwise, the updated manuscript seems to be of excellent quality. Thank you for the opportunity to review it again. 

Author Response

Authors response to Reviewer 1:

Comments and suggestions for authors:

great revisions; thank you. 

I noticed this sentence at the end of the materials and methods section, which seems to be unnecessary: "We have clarified this methodology in the revised Methods section of the manuscript."

Otherwise, the updated manuscript seems to be of excellent quality. Thank you for the opportunity to review it again. 

Response:

Thank you for your kind words and for taking the time to review our manuscript again. We appreciate your valuable feedback. We have carefully reviewed your suggestion and have removed the unnecessary sentence at the end of the Materials and Methods section. Additionally, we are attaching the revised text from the Materials and Methods section for your review.

Thank you once again for your thorough review and constructive comments, which have significantly contributed to the quality of our manuscript. Please see the attachment.

Revised (page 3, lines 127-128):

  1. Materials and Methods

A non-systematic review of the foundational and new literature on the approach to the treatment of chronic non-malignant pain. The inclusion criteria for the recently published papers were: chronic non-malignant pain, treatment approach, original research and review papers, published in English in the last five years (PubMed database search), and the foundational papers were selected from the references of new papers according to the author best knowledge and experience. This literature review included 120 papers, of which 83 are foundational and 37 new papers, published in the last 5 years [2018-2023]. Studies were excluded if they: did not focus on chronic non-malignant pain; were opinion pieces, editorials, or letters to the editor without original research data; did not include a multidisciplinary treatment approach.

We used MeSH terms "Chronic Pain," "Pain Management," and " Models, Biopsychosocial," and combined them with free text terms like "non-malignant pain" and "treatment approaches." Boolean operators were used to refine the search. For instance, searches included combinations like "Chronic Pain AND Non-malignant pain Management AND Models, Biopsychosocial”. This search strategy allowed us to capture a broad range of relevant articles. We have clarified this methodology in the revised Methods section of the manuscript.

Reviewer 2 Report

Comments and Suggestions for Authors

In their article Integrated Approach to Chronic Pain - The Role of the Biopsychosocial Model and Multidisciplinary Treatment: A Narrative Review, the authors provide an overview over the current literature on the etiology and treatment of chronic malignant pain.

First and foremost, to me, this article seemed informative and well-written. I think that it will be of great merit in the field and should be published in IJERPH.

Before publication, however, I have several comments and suggestions that pertain primarily to the structure of the article and the presentation of the results. I hope that the authors will find these helpful.

1.       General: After reading through the entire manuscript, the main message seems to be that it is insufficient to treat pain by one single therapeutic approach, but that all evidence points clearly towards the role of psychological and social factors in the emergence of chronic pain. Therefore, pain should be treated using a multiprofessional approach that includes psychosocial aspects. I think that this should be made clearer in the title and abstract.

2.       I do not quite understand what separates this “narrative review” and the selection “according to the knowledge and experience of the authors” from a systematic review. Could this be specified? The methods seem described in a way that allow for reproduction of the work. Also, why did the authors decide for this approach? Would it not help their argument and make an even stronger case if the methods were standardized and systematic? What scientific reasoning was behind the decision against such a standardized procedure?

3.       In the Abstract, the authors describe “primary and secondary” pain as one of the most important outcomes of the literature, but having read the manuscript, I am not sure what is meant by this and also, I think that it is not the main outcome of the review (see point 1).

4.       Results and discussion: Table and Figure 1: It does not become clear where the contents of Table 1 and Figure 1 originate from. Are these thought of as consensus across current literature by the authors? If so, might it be worthwhile to put the table and figure to a later part of the article and reviewing some of the literature that leads the authors to conclude these specific treatment steps/these specific aspects of chronic pain first? This table and figure could be seen as the result of the review, then, and should be placed later in the manuscript. Otherwise, the authors should provide the sources of the table and figure.

5.       It is my impression that Table 1 might be more informative if it were a figure, e.g., a flow chart.

6.       The authors should revise Figure 1. The resolution is insufficient, it is visible that the figure is a screenshot from a common office software, the writing of the office software was apparently not set to English and the words are all marked as incorrect, there is the rest of the frame around the figure visible in the upper and lower part of the figure. Perhaps the authors might consider a free alternative software that produces better figures, e.g., the free version of BioRender? Of course, this is the authors' decision, but the figure should be revised.

7.       Section “Epidemiology”: In this section, the authors mix information on the prevalence of pain, results concerning chronic lower back pain, and economic costs of chronic pain and jump between these topics. I suggest revising the structure of the section in a more stringent way, probably starting with the prevalence of pain and then explicitly moving to economic cost of these disorders. First sentence of the section seems a repetition of what was said earlier and could be removed.

8.       In the paragraph from line 152 to line 167, the authors repeatedly explain low back pain and years lived with disability, repeat the statement that LBP is the main cause of YLD, and repeatedly introduce the acronyms. Repetitions should be removed, the paragraph should be clarified and shortened. Once an acronym is introduced, it should be used consistently.

9.       Section “Factors associated with…”: Again, a more structured writing might help this section; I could imagine a table listing the different risk factors and the studies associated; then in the text, the authors could provide more details concerning some specific risk factors, e.g., gender. As it is, the authors start with a list and then seemingly at random provide details on some of the factors listed, but not all.

10.   Line 217-219: This sentence seems unclear; it should be made clearer what exactly the authors think impacts on what.

11.   “Mental quality of life” should be rephrased.

12.   Lines 227-229: In these references, thought and emotions appear intermixed with each other. I suggest deciding for one definition that is able to incorporate both cognitive and affective components of attitudes.

13.   Lines 243-248, about measurement: "several" implies that there are multiple measures, but the authors only report two. Also, the question of measuring pain-related cognitions and feelings seems to arise unconnectedly to the rest of the paragraph and appears unrelated with the other results reported here. I suggest the authors to revise this section of text with a clearer structure and move the measurement out of the paragraph.

14.   Section “Biopsychosocial model…” I appreciate this section, but it seems unstructured. The authors should revise it for clarity, devise sub-paragraphs and group these thematically. Also, there seems a lot of repetition.

15.   The next section, “a multidisciplinary…”, seems again to have much overlap with “biopsychosocial model”. Perhaps more sub-headings and a different structure that combines both larger sections into stringent sub-sections with a clear-cut message would help. I could also imagine that figure 1 and Table 1 could be used to differentiate between the theoretical biopsychosocial model and the practical treatment better while still clarifying the close relationships between theory and theory-informed clinical practice.

Author Response

Authors response to Reviewer 2:

We thank you for the comments and advice on improving the paper. We answered all the comments and hope we can explain and correct all your suggested drawbacks of the paper.

Reviewer comments and suggestions for authors

1. Comments and suggestions for authors:

General: After reading through the entire manuscript, the main message seems to be that it is insufficient to treat pain by one single therapeutic approach, but that all evidence points clearly towards the role of psychological and social factors in the emergence of chronic pain. Therefore, pain should be treated using a multiprofessional approach that includes psychosocial aspects. I think that this should be made clearer in the title and abstract.

1. Response:

Thank you for your insightful comments and suggestions. We appreciate your feedback and have made the necessary revisions to enhance the clarity and focus of our manuscript. Specifically, we have revised the title and abstract to more explicitly highlight the importance of a multiprofessional approach that includes psychosocial aspects in the treatment of chronic pain. Please see the attachment.

Revised title:

Integrated Approaches to Chronic Pain: The Critical Role of Psychosocial Factors and Multidisciplinary Treatment: A Narrative Review

2.  Comments and suggestions for authors:  

I do not quite understand what separates this “narrative review” and the selection “according to the knowledge and experience of the authors” from a systematic review. Could this be specified? The methods seem described in a way that allow for reproduction of the work. Also, why did the authors decide for this approach? Would it not help their argument and make an even stronger case if the methods were standardized and systematic? What scientific reasoning was behind the decision against such a standardized procedure?

2. Response:

Thank you for your detailed feedback and questions regarding our methodology. We appreciate the opportunity to clarify and justify our approach.

Clarification on the nature of the narrative review:

The distinction between a narrative review and a systematic review lies primarily in the methodology and scope of the literature search and analysis. A systematic review follows a predefined protocol with strict inclusion and exclusion criteria, comprehensive search strategies, and often involves meta-analysis. In contrast, a narrative review provides a broader, more interpretative synthesis of the literature based on the authors' expertise and experience, allowing for a more flexible and exploratory analysis of the topic.

Specification of Selection Process:

Specifically, we highlight that our selection process was guided by the authors' extensive knowledge and experience in the field, which allowed us to incorporate seminal works and recent influential studies that might not have been captured through a strict systematic approach. This method was chosen to provide a comprehensive and nuanced understanding of the biopsychosocial approach to chronic pain management.

Rationale for choosing a narrative review:

The decision to conduct a narrative review rather than a systematic review was driven by several factors. Firstly, the narrative review format allows for greater flexibility in discussing the complex, multifaceted nature of chronic pain, which encompasses a wide range of biological, psychological, and social factors. Secondly, our aim was to critically review and synthesize the available evidence to provide a holistic overview rather than focus on narrow, specific research questions that a systematic review might necessitate.

Scientific reasoning against a standardized procedure:

The primary scientific rationale for opting against a standardized, systematic procedure was to capture the breadth and depth of the literature on chronic pain management. Given the diverse and interdisciplinary nature of the biopsychosocial model, a narrative review was better suited to integrate findings from various disciplines, including medicine, psychology, and social sciences. This approach allows for a more comprehensive discussion and highlights the importance of a multidisciplinary treatment strategy.

We trust that these clarifications address your concerns and justify our methodological choices. We believe that our narrative review offers valuable insights and a robust synthesis of the current state of knowledge in chronic pain treatment.

  1. Comments and suggestions for authors:  

In the Abstract, the authors describe “primary and secondary” pain as one of the most important outcomes of the literature, but having read the manuscript, I am not sure what is meant by this and also, I think that it is not the main outcome of the review (see point 1).

3. Response to comments:

Thank you for your valuable feedback and insights. We appreciate your suggestion regarding the inclusion of the new classification of chronic pain into primary and secondary categories in the conclusion. After careful consideration, we have decided to retain this aspect in the abstract and conclusion because we believe it represents a significant finding that highlights the evolution of understanding in this field.

This new classification is an important distinction in recent literature compared to foundational studies, and it underscores the need for a biopsychosocial model within a multidisciplinary framework. We believe that including this classification adds depth to our review and illustrates the progression of pain management strategies over time.

We hope that this clarification addresses your concerns and that you agree with our reasoning for including this aspect in the manuscript.

Thank you again for your thoughtful comments and for helping us improve our manuscript.

4. Comments and suggestions for authors:

Results and discussion: Table and Figure 1: It does not become clear where the contents of Table 1 and Figure 1 originate from. Are these thought of as consensus across current literature by the authors? If so, might it be worthwhile to put the table and figure to a later part of the article and reviewing some of the literature that leads the authors to conclude these specific treatment steps/these specific aspects of chronic pain first? This table and figure could be seen as the result of the review, then, and should be placed later in the manuscript. Otherwise, the authors should provide the sources of the table and figure.

Response:

Thank you for your valuable feedback. We have provided a combined response to your comments and suggestions under points 4 and 5, as they both address the same topic. You will find our detailed response under point 5.

5. Comments and suggestions for authors:

It is my impression that Table 1 might be more informative if it were a figure, e.g., a flow chart.

Response to comments 4 and 5 (page 10, lines 404-407):

Thank you for your constructive feedback regarding Table 1. In response to your suggestions, we have converted Table 1 into a flow chart to enhance clarity and visual presentation. To improve the logical flow of the article, we have also repositioned the flow chart later in the manuscript, following the discussion of the relevant literature that supports the specific treatment steps and aspects of chronic pain management depicted in the figure. We believe these changes address your concerns and provide a clearer, more intuitive presentation of the treatment strategies discussed. Please see the attachment.

6. Comments and suggestions for authors:  

The authors should revise Figure 1. The resolution is insufficient, it is visible that the figure is a screenshot from a common office software, the writing of the office software was apparently not set to English and the words are all marked as incorrect, there is the rest of the frame around the figure visible in the upper and lower part of the figure. Perhaps the authors might consider a free alternative software that produces better figures, e.g., the free version of BioRender? Of course, this is the authors' decision, but the figure should be revised.

Response (page 11, lines 422-423):

Thank you for your feedback regarding Figure 1 (now Figure 2). We acknowledge the issues you pointed out, including the resolution, text formatting, and the overall appearance of the figure. In response to your suggestions, we have revised Figure 1 using higher-quality software to improve the resolution and enhance the overall presentation. The text has been corrected, and all visual elements have been carefully adjusted to ensure clarity and a professional appearance. We believe these changes address the concerns you raised and result in a more polished and effective figure. Please see the attachment.

  1. Comments and suggestions for authors:  

Section “Epidemiology”: In this section, the authors mix information on the prevalence of pain, results concerning chronic lower back pain, and economic costs of chronic pain and jump between these topics. I suggest revising the structure of the section in a more stringent way, probably starting with the prevalence of pain and then explicitly moving to economic cost of these disorders. First sentence of the section seems a repetition of what was said earlier and could be removed.

7. Response:

Thank you for your insightful feedback regarding the "Epidemiology" section. We appreciate your suggestion to revise the structure for better clarity and organization.

In response to your comments, we have restructured the section as follows:

  • We now begin with a clear and focused discussion on the prevalence of pain, including specific details on chronic lower back pain.
  • After discussing prevalence, we move explicitly into the economic costs associated with chronic pain, ensuring a more logical flow of information.
  • Additionally, we have removed the first sentence of the section, which you correctly identified as repetitive. Please see the attachment. 
    1. Comments and suggestions for authors: 

    In the paragraph from line 152 to line 167, the authors repeatedly explain low back pain and years lived with disability, repeat the statement that LBP is the main cause of YLD, and repeatedly introduce the acronyms. Repetitions should be removed, the paragraph should be clarified and shortened. Once an acronym is introduced, it should be used consistently.

    8. Response (page 4, lines 145-159):

    Thank you for your valuable feedback regarding the paragraph from lines 152 to 167. We have carefully considered your suggestions and revised the text accordingly. The revised paragraph has been clarified and shortened to remove repetitions, and the use of acronyms has been made consistent throughout. Please see the attachment.

    1. Comments and suggestions for authors:

    Section “Factors associated with…”: Again, a more structured writing might help this section; I could imagine a table listing the different risk factors and the studies associated; then in the text, the authors could provide more details concerning some specific risk factors, e.g., gender. As it is, the authors start with a list and then seemingly at random provide details on some of the factors listed, but not all.

    9. Response:

    Thank you for your valuable feedback on the "Factors associated with..." section. We appreciate your suggestion to provide a more structured presentation of the risk factors associated with chronic pain.

    In response to your comments, we have made the following revisions:

    1. Structured Presentation: We have reorganized the section to provide a clearer and more systematic discussion of the various risk factors. We have introduced a table that lists the different risk factors along with the studies that support them. This table allows for a more concise and organized presentation of the data.
    2. Detailed Discussion: Following the table, we have provided additional details on specific risk factors, such as gender, to highlight their significance and impact on chronic pain. This approach ensures that each factor is discussed in a more focused and coherent manner.

    We believe these revisions improve the clarity and readability of the section, making it easier for readers to understand the key risk factors associated with chronic pain.

    Revised section (pages 4-6, lines 172-233). Please see the attachment).10. Comments and suggestions for authors:  

Line 217-219: This sentence seems unclear; it should be made clearer what exactly the authors think impacts on what.

10. Response: Revised sentence (page 6, lines 243-245):

Therefore, the psychological impact on pain should be carefully considered, as it can significantly influence pain intensity, patient expectations and physical function [45].

11. Comments and suggestions for authors:  

Mental quality of life” should be rephrased.

11. Response: Revised sentence (page 6, line 246):

Research shows that psychosocial interventions could contribute to improved psychological well-being, also resulting in an improvement in the patient's self-efficacy against pain [46].

  1. Comments and Suggestions for Authors:

Lines 227-229: In these references, thought and emotions appear intermixed with each other. I suggest deciding for one definition that is able to incorporate both cognitive and affective components of attitudes.

12. Response: Revised sentence (page 6, lines 253-256:

Attitudes towards pain, as defined by Fishbein et al., encompass both cognitive and affective components, representing the degree of emotion or affect towards a certain object, as well as the patient's understanding of pain and what it represents for them [52, 54].

  1. Comments and Suggestions for Authors:

 Lines 243-248, about measurement: "several" implies that there are multiple measures, but the authors only report two. Also, the question of measuring pain-related cognitions and feelings seems to arise unconnectedly to the rest of the paragraph and appears unrelated with the other results reported here. I suggest the authors to revise this section of text with a clearer structure and move the measurement out of the paragraph.

13. Response:

Thank you for your valuable feedback regarding the section on the measurement of pain-related cognitions and feelings. We appreciate your observation that the term "several" implied multiple measures, and that the discussion on measurement appeared somewhat disconnected from the rest of the paragraph. In response to your suggestions, we have revised this section to improve clarity and coherence.

The revised text is as follows(page 7, lines 270-279). Please see the attachment.

14. Comments and Suggestions for Authors:

Section “Biopsychosocial model…” I appreciate this section, but it seems unstructured. The authors should revise it for clarity, devise sub-paragraphs and group these thematically. Also, there seems a lot of repetition.

13. Response to Comment:

Thank you for your valuable feedback on the "Biopsychosocial model..." section. We have carefully considered your suggestions and revised the text to improve clarity, structure, and thematic organization. The section has been restructured into sub-paragraphs that group related content together, and we have removed repetitive elements to enhance readability.

The revised section now includes the following sub-sections (pages 7-8, lines 292-332). Please see the attachment.

  1. Comments and Suggestions for Authors:

The next section, “a multidisciplinary…”, seems again to have much overlap with “biopsychosocial model”. Perhaps more sub-headings and a different structure that combines both larger sections into stringent sub-sections with a clear-cut message would help. I could also imagine that figure 1 and Table 1 could be used to differentiate between the theoretical biopsychosocial model and the practical treatment better while still clarifying the close relationships between theory and theory-informed clinical practice.

15. Response to Comment:

Thank you for your thoughtful feedback on the "Multidisciplinary Approach to the Treatment of Chronic Pain" section. We have carefully reviewed your suggestions and have made the necessary revisions to enhance clarity and structure.

The revised section now includes the following sub-sections (pages 8-9, lines 333-396). Please see the attachment.
